



# Snow cover duration trends observed at sites and predicted by multiple models

Richard Essery[1], Hyungjun Kim[2], Libo Wang[3], Paul Bartlett[3], Aaron Boone[4], Claire Brutel-Vuilmet[5], Eleanor Burke[6], Matthias Cuntz[7], Bertrand Decharme[4], Emanuel Dutra[8], Xing Fang[9], Yeugeniy Gusev[10], Stefan Hagemann[11], Vanessa Haverd[12], Anna Kontu[13], Gerhard Krinner[5], Matthieu Lafaysse[14], Yves Lejeune[14], Thomas Marke[15], Danny Marks[16], Christoph Marty[17], Cecile B. Menard[1], Olga Nasonova[10], Tomoko Nitta[2], John Pomeroy[9], Gerd Schädler[18], Vladimir Semenov[19], Tatiana Smirnova[20], Sean Swenson[21], Dmitry Turkov[22], Nander Wever[17,23], and Hua Yuan[24]

[1]School of GeoSciences, University of Edinburgh, Edinburgh, UK
[2]Institute of Industrial Science, University of Tokyo, Tokyo, Japan
[3]Climate Research Division, Environment and Climate Change Canada, Toronto, Canada
[4]Université de Toulouse, Météo-France, CNRS, Toulouse, France
[5]CNRS, Université Grenoble Alpes, Institut de Géosciences de l'Environnement, Grenoble, France
[6]Met Office, Exeter, UK
[7]Université de Lorraine, AgroParisTech, INRAE, UMR Silva, Nancy, France
[8]Instituto Dom Luiz, Faculdade de Ciências, Universidade de Lisboa, Lisbon, Portugal
[9]Centre for Hydrology, University of Saskatchewan, Saskatoon, Canada
[10]Institute of Water Problems, Russian Academy of Sciences, Moscow, Russia
[11]Institute of Coastal Research, Helmholtz-Zentrum Geesthacht, Geesthacht, Germany
[12]CSIRO Oceans and Atmosphere, Canberra, ACT, Australia
[13]Space and Earth Observation Centre, Finnish Meteorological Institute, Sodankylä, Finland
[14]Météo-France, CNRS, CNRM, Centre d'Etudes de la Neige, Grenoble, France
[15]Department of Geography, University of Innsbruck, Innsbruck, Austria
[16]USDA Agricultural Research Service, Boise, ID, USA
[17]WSL Institute for Snow and Avalanche Research SLF, Davos, Switzerland
[18]Institute of Meteorology and Climate Research, Karlsruhe Institute of Technology, Karlsruhe, Germany
[19]A.M. Obukhov Institute of Atmospheric Physics, Russian Academy of Sciences, Moscow, Russia
[20]Cooperative Institute for Research in Environmental Science/Earth System Research Laboratory, NOAA, Boulder, CO, USA
[21]Climate and Global Dynamics Division, National Center for Atmospheric Research, Boulder, CO, USA
[22]Institute of Geography, Russian Academy of Sciences, Moscow, Russia
[23]Department of Atmospheric and Oceanic Sciences, University of Colorado, Boulder, CO, USA
[24]School of Atmospheric Sciences, Sun Yat-sen University, Guangzhou, China

**Correspondence:** Richard Essery (richard.essery@ed.ac.uk)

**Abstract.** Thirty-year simulations of seasonal snow cover in 22 physically based models driven with bias-corrected meteorological reanalyses are examined at four sites with long records of snow observations. Annual snow cover durations differ widely between models but interannual variations are strongly correlated because of the common driving data. No significant trends are observed in starting dates for seasonal snow cover, but there are significant trends towards snow cover ending earlier at two of the sites in observations and most of the models. A simplified model with just two parameters controlling solar radiation and sensible heat contributions to snowmelt spans the ranges of snow cover durations and trends. This model predicts that



sites where snow persists beyond annual peaks in solar radiation and air temperature will experience rapid decreases in snow cover duration with warming as snow begins to melt earlier and at times of year with more energy available for melting.

# 1  Introduction

The extensive seasonal snow cover of Northern Hemisphere land is sensitive to climate warming and strongly influences surface-atmosphere interactions, so it is important that climate models should be able to simulate it accurately. Observations of snow cover extent have been used to demonstrate climate change and to evaluate climate models in all five Intergovernmental Panel on Climate Change (IPCC) Working Group 1 Assessment Reports to date. Reports from the second onwards have demonstrated a strong relationship between decreasing snow cover and increasing air temperature in observations, and have

drawn on multi-model simulations in the Atmospheric Model Intercomparison Project (AMIP) and the Coupled Model Intercomparison Project (CMIP) coordinated by the World Climate Research Programme. Although the reproduction of seasonal snow cover by climate models has improved, CMIP5 simulations underestimated significant reductions observed in spring snow cover extent (Brutel-Vuilmet et al., 2013) and had a wide spread in predictions of snow-albedo feedback strength (Qu and Hall, 2014). In preparation for the sixth IPCC assessment report, climate modelling centres have now performed CMIP6

coupled land-atmosphere-ocean simulations with their latest models. Mudryk et al. (2020) report an overall better representation of Northern Hemisphere snow cover extent in the CMIP6 multi-model ensemble than in CMIP5, but a large spread remains in simulated trends.

In addition to coupled model experiments, snow simulations by stand-alone land surface models have been driven with prescribed meteorological variables on global grids in the Global Soil Wetness Project (Dirmeyer et al., 2006) and at individual

sites in the Project for Intercomparison of Land-surface Parameterization Schemes (Slater et al., 2001) and the Snow Model Intercomparison Project (Etchevers et al., 2004; Essery et al., 2009). These studies have invariably found wide ranges in simulations and inconsistencies in model performance. The Earth System Model-Snow Model Intercomparison Project (ESM-SnowMIP; Krinner et al., 2018) includes simulations driven with both in situ meteorological measurements and bias-corrected reanalyses at ten well-instrumented snow study sites; simulations were first evaluated with between seven and twenty years

of in situ driving data, but using reanalyses allows longer simulations for investigating trends. This paper examines observed trends in seasonal snow cover duration and simulations driven with 1980-2010 bias-corrected reanalyses at four of the ESM-SnowMIP sites selected because they had at least 27 years of daily snow observations up to 2010. The locations of the sites are given in Table 1. Reflecting motivations for the establishment of snow study sites by national organizations, Col de Porte (France), Reynolds Mountain East (USA) and Weissfluhjoch (Switzerland) are at high elevations in mid-latitude mountains,

whereas Sodankylä (Finland) is a low elevation Arctic site. All of the sites typically have between five and eight months of continuous winter snow cover and can have shorter periods of ephemeral snow cover at other times of year.

Simple empirical models of snowmelt are still often used for hydrological and glaciological applications, but all of the models participating in ESM-SnowMIP are physically based and calculate coupled mass and energy balances for snow on the ground. Eighteen groups submitted simulations by 22 models and model variants driven with a common set of bias-corrected



**Table 1.** Site locations and 0.5° grid elevations

| Site | Latitude | Longitude | Elevation | Grid elevation |
|------|----------|-----------|-----------|----------------|
| Col de Porte | 45.30° N | 5.77° E | 1325 m | 870 m |
| Reynolds Mountain East | 43.19° N | 116.78° W | 2060 m | 1260 m |
| Sodankylä | 67.37° N | 26.63° E | 179 m | 220 m |
| Weissfluhjoch | 46.83° N | 9.81° E | 2536 m | 1930 m |

reanalyses provided by the third Global Soil Wetness Project (GSWP3) for the Land Surface, Snow and Soil moisture Model Intercomparison Project (van den Hurk et al., 2016). The models include land surface schemes that are commonly coupled to atmosphere models (CABLE, CLASS, CLM5, CoLM, EC-Earth, ISBA, MATSIRO, RUC, two versions of JSBACH, three configurations of JULES and two versions of ORCHIDEE), stand-alone land surface or hydrology models (CRHM, ESCIMO, SPONSOR, SWAP and Veg3D), and snow physics models (Crocus and SNOWPACK); references for all of these models can

be found in Table 1 of Krinner et al. (2018). Although snow models are much less complex than comprehensive Earth System Models, they have sufficient complexity and large enough parameter spaces to make it difficult to interpret why they behave in the ways that they do. For Earth System Models, Randall et al. (2019) concluded that "we must work to create much simpler models that can semiquantitatively reproduce the key results of the comprehensive models". In that spirit, a highly simplified two-parameter energy balance model ("2PM" hereafter) is used to interpret the results of the ESM-SnowMIP models.

## 2   Methods


All of the meteorological variables required to drive physically based mass and energy balance snow models (air temperature, humidity and pressure, snowfall and rainfall rates, shortwave and longwave radiation fluxes, and wind speed) for 1980-2010 at the ESM-SnowMIP sites were extracted from the GSWP3 dataset and interpolated from three-hourly to hourly timesteps. Because coupling to an atmosphere model was not required, snow models that are not part of an Earth System Model were

also able to participate in this component of ESM-SnowMIP. For GSWP3, the 20th Century Reanalysis was used to nudge the dynamics of the Global Spectral Model for downscaling from 2° to 0.5° resolution. Biases in monthly means of temperature, diurnal temperature range, precipitation and radiation fluxes relative to Climate Research Unit Time-Series (CRUTS), Global Precipitation Climatology Centre and Surface Radiation Budget datasets were then removed. Additional bias corrections had to be applied for ESM-SnowMIP site simulations because the mountain sites are at much higher elevations than the 0.5° GSWP3

grid cells in which they lie (Table 1). Biases relative to in situ measurements for overlapping periods at each site were simply removed for all driving variables, thus preserving distribution shapes, seasonal cycles and trends from the GSWP3 dataset (Menard et al., 2019). The meteorological variables extracted from GSWP3, interpolated to hours and bias-corrected to the sites are referred to as the driving data for the ESM-SnowMIP models hereafter.



The simplified model that will be used for interpreting the ESM-SnowMIP results below has two fixed dimensionless parameters: a snow albedo $\alpha$ and a surface-atmosphere turbulent exchange coefficient $C_H$. A large simplification comes from neglecting heat required to warm snow to the melting point in comparison with heat required to melt snow. Snowmelt rate $M$ (kg m$^{-2}$ s$^{-1}$) is predicted by the energy balance equation

$$\lambda_m M = (1-\alpha)SW_\downarrow + LW_\downarrow - \sigma T_s^4 - H - \lambda_s E \tag{1}$$

with latent heat of melting $\lambda_m$ ($0.334\times10^6$ J kg$^{-1}$), latent heat of sublimation $\lambda_s$ ($2.835\times10^6$ J kg$^{-1}$), surface temperature $T_s$ (K) and Stefan-Boltzmann constant $\sigma$ ($5.67\times10^{-8}$ W m$^{-2}$ K$^{-4}$); $SW_\downarrow$ and $LW_\downarrow$ (W m$^{-2}$) are downward shortwave and longwave radiation fluxes, and heat advected by rain falling on snow is neglected. Sensible heat flux $H$ (W m$^{-2}$) and moisture flux $E$ (kg m$^{-2}$ s$^{-1}$) from the surface to the atmosphere are parametrized using the bulk formulae

$$H = \rho c_p C_H U(T_s - T_a) \tag{2}$$

and

$$E = \rho C_H U[q_{\text{sat}}(T_s, p) - q_a] \tag{3}$$

for air pressure $p$ (Pa), temperature $T_a$, specific humidity $q_a$, heat capacity $c_p$ (1005 J K$^{-1}$ kg$^{-1}$) and density $\rho$ (kg m$^{-3}$); $U$ (m s$^{-1}$) is wind speed and $q_{\text{sat}}$ is the specific humidity of saturated air. Equations (1) to (3) are first solved for unknown $T_s$ with $M = 0$. If this gives a temperature greater than $T_m = 273.15$ K while there is snow on the ground, the equations are solved again for unknown $M$ with $T_s = T_m$. Melt and sublimation rates are then used with snowfall rate $S_f$ (kg m$^{-2}$ s$^{-1}$) each hour in the mass balance equation

$$\frac{dS}{dt} = S_f - E - M \tag{4}$$

to predict changes in snow mass $S$ (kg m$^{-2}$), which is limited to be greater than or equal to zero and is converted to depth using a fixed snow density of 300 kg m$^{-3}$.

## 3 Results

Figure 1 shows monthly means and trends in air temperatures measured at the sites and in the driving data; averages and ranges of observed start and end dates for continuous seasonal snow cover are also shown. The seasonal temperature cycle and trends in the driving data match observations closely at Sodankylä because the station there was included in the CRUTS database used for correcting GSWP3 temperatures. Weissfluhjoch is 60 km from the nearest CRUTS station at Säntis but only 50 m higher. Col de Porte is 75 km from Lyon and 1125 m higher, and Reynolds Mountain East is 65 km from Boise and 1190 m higher, but temperature trends in the driving data are still similar to observations, particularly for significant trends (i.e. when 95% confidence intervals do not cross zero). The driving data have significant 1980-2010 warming trends for April to June at Col de Porte, July and September at Reynolds Mountain East, August and December at Sodankylä, and June at Weissfluhjoch. The





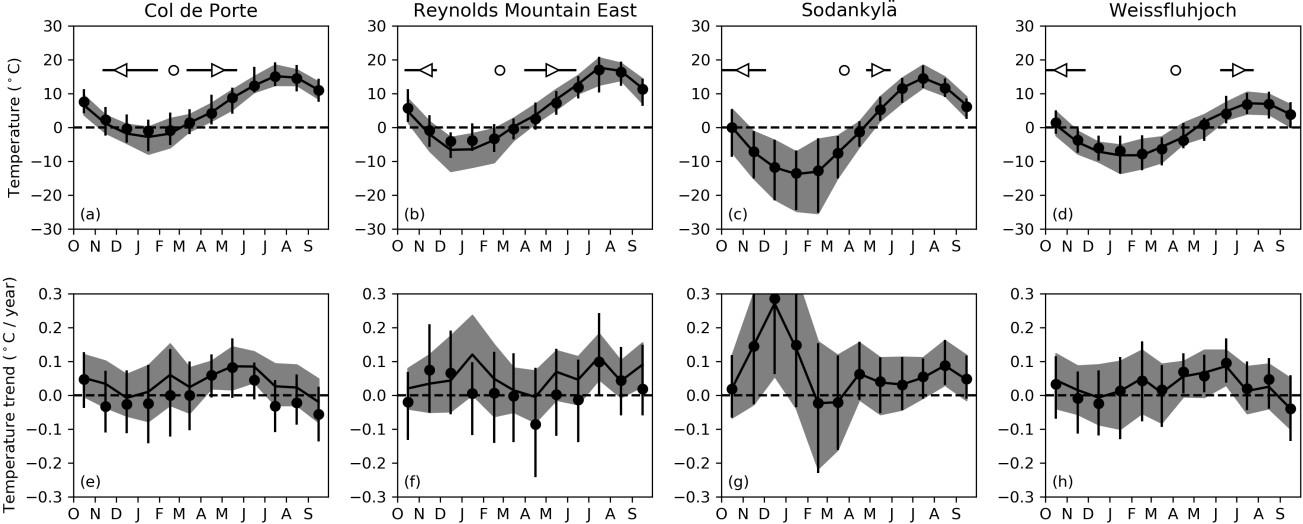

**Figure 1.** (a-d): Monthly-mean temperatures calculated from measurements at the sites (filled circles) and the driving data (lines). Vertical bars for measurements and grey bands for driving data show ranges between the warmest and coolest months from October 1980 to September 2010. Triangles and horizontal bars show averages and ranges of observed start (◁) and end (▷) dates of continuous seasonal snow cover. Open circles show average dates of maximum snow depth. (e-h): Monthly temperature trends calculated by the Theil-Sen method (Sen, 1968). Vertical bars for measurements and grey bands for driving data show 95% confidence intervals.

rapid December warming at Sodankylä will not directly influence simulated snow cover durations because it corresponds with a reduction in the occurrence of very low temperatures at times when snow is not melting. Other warming trends at Reynolds

Mountain East and Sodankylä occur during snow-free months, but warming trends at Col de Porte and Weissfluhjoch overlap the normal periods of snowmelt.

Solid precipitation is notoriously difficult to measure accurately, and quality-controlled measurements of snowfall are not available for all years back to 1980 at all of the sites. Annual snowfall amounts derived from precipitation gauge measurements are therefore only shown for comparison with the driving data in Figure 2, and snowfall trends will only be investigated in the

driving data. Weissfluhjoch is the only site with a significant downward trend in snowfall at the 95% confidence level, although Col de Porte has a downward trend with a 90% confidence interval from -15 to -0.4 mm per year. In contrast with the lack of trend at Reynolds Mountain East, Nayak et al. (2010) found significant decreases in the fractions of annual precipitation falling as snow at lower elevations in the Reynolds Creek Experimental Watershed. Sodankylä had higher snowfall in the 1990s than in the 1980s and 2000s in both the site measurements and the driving data, and no overall snowfall trend from 1980 to

2010. Irannezhad et al. (2016), however, found a significant decreasing winter precipitation trend at Sodankylä in a longer series of measurements from 1909 to 2008. Figures 1 and 2 together show that Sodankylä has the lowest winter temperatures and the lowest snowfall of the sites; Col de Porte has the warmest winter temperatures and the shortest seasonal snow cover; Weissfluhjoch has the highest snowfall, coolest summer temperatures and longest seasonal snow cover.





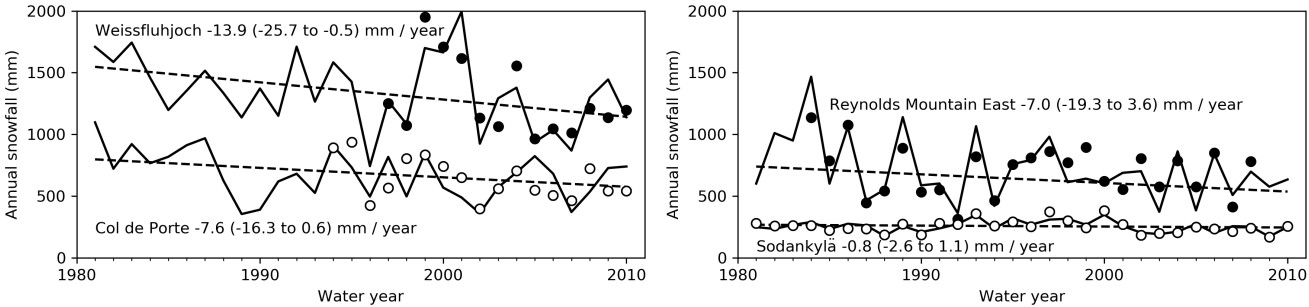

**Figure 2.** Water equivalent snowfall calculated from measurements at the sites (open and filled circles) and the driving data (solid lines) for water years starting on 1 October. Driving data trends are given with 95% confidence intervals in parentheses.

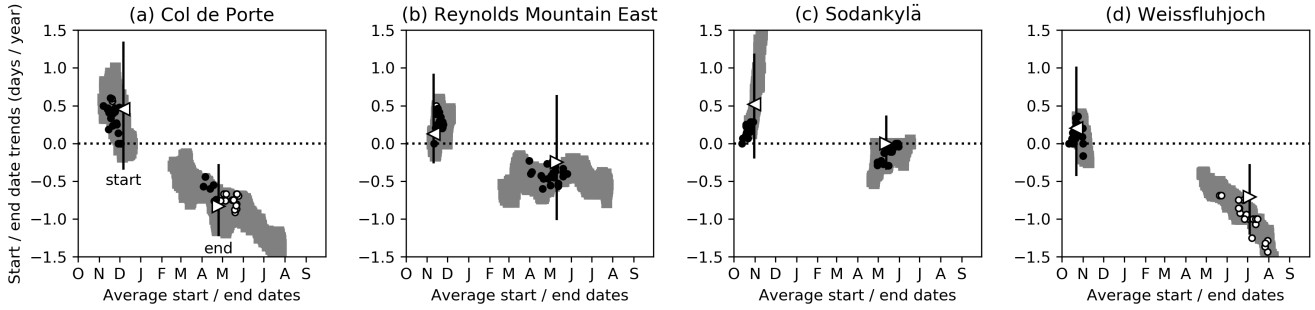

**Figure 3.** Scatter plots of averages and trends in start and end dates of continuous seasonal snow cover observed at the sites (triangles) and predicted by the ESM-SnowMIP models (open circles for significant trends, filled circles for insignificant trends) and 2PM (grey). Vertical bars show 95% confidence intervals on observed start (◁) and end (▷) date trends.

Start and end dates for continuous seasonal snow cover were found by searching for the first and last dates with snow depths

exceeding 2 cm before and after the dates of maximum snow depth in each year. Figure 3 shows averages and trends for start and end dates in observations and simulations at all of the sites (annual time series from which these were calculated are shown in supplementary Figure A1). Simulated start dates are largely determined by snowfall in the driving data and show relatively little spread between the models, except that some models will melt early snowfall at Col de Porte and others will retain it on the ground. Trends towards later start dates are observed at all sites and in most model simulations, but none of

these trends are found to be significant with 95% confidence. Simulated end dates are influenced by differences in how models respond to increasing air temperatures and solar radiation in spring, leading to larger spreads between models. The spread is particularly large for Weissfluhjoch; two of the models melt snow consistently earlier than the others, and three models retain year-round snow cover in some years (which has never been observed in measurements going back to 1936 at Weissfluhjoch). Years in which a model does not melt the snow are excluded from calculations of end dates. Significant trends towards earlier

snow disappearance are observed at Col de Porte and Weissfluhjoch but not at Reynolds Mountain East or Sodankylä, and





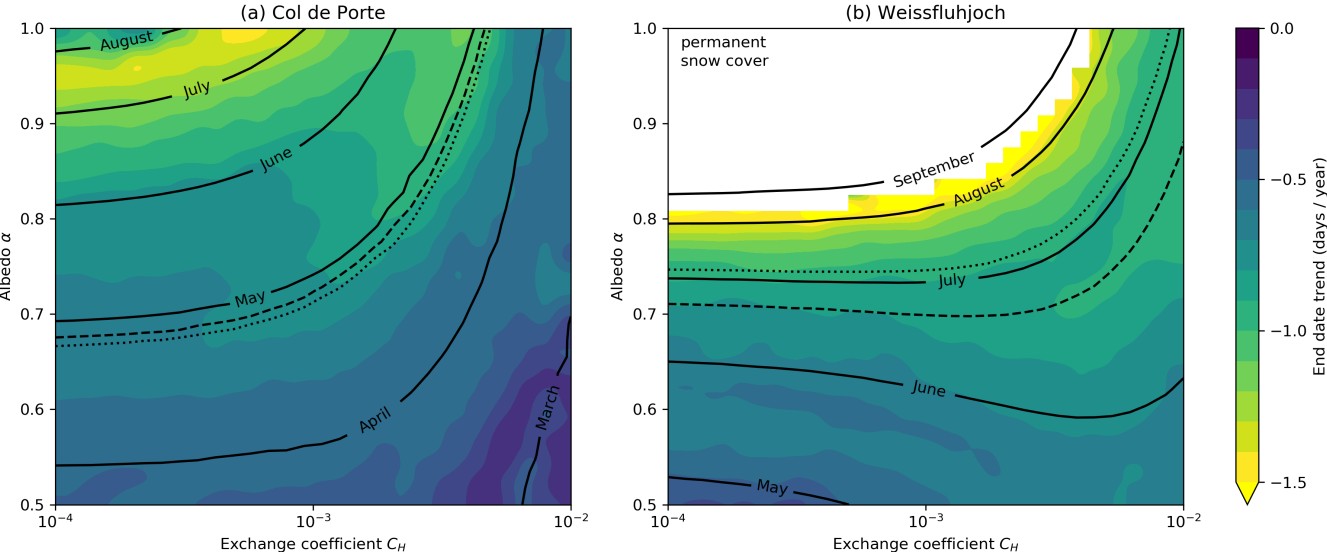

**Figure 4.** Averages (solid contours) and trends (colours) for continuous snow cover end dates in 2PM simulations at (a) Col de Porte and (b) Weissfluhjoch, excluding simulations with permanent snow cover. Dotted lines are contours for average observed snow cover end dates and dashed lines are dates of maximum warming trends in the driving data (Figure 1).

most models lie within the confidence intervals of the observed trends. Fifteen of the 22 Col de Porte simulations and all of the Weissfluhjoch simulations have significant trends. Reductions in snow cover have previously been detected using the same observations at Col de Porte by Lejeune et al. (2019) and at Weissfluhjoch by Marty and Meister (2012). The remaining discussion here will focus on model behaviour at those two sites.

2PM was run 10,000 times for each site with snow albedos ranging from 0.5 to 1 and turbulent exchange coefficients ranging from $10^{-4}$ to $10^{-2}$; these unrealistically wide parameter ranges give results that encompass and extend beyond the ESM-SnowMIP model results in Figure 3. Increasingly negative trends in continuous snow cover end dates for 2PM simulations that melt snow later in the year at Col de Porte and Weissfluhjoch are striking features of Figure 3. The same behaviour is seen most clearly for Weissfluhjoch in the ESM-SnowMIP models.

Snow albedo and turbulent exchanges between the surface and the atmosphere vary with time in reality and in realistic models, but 2PM results can be plotted as contours or a colour scale on the fixed $\alpha - C_H$ parameter space. Figure 4 overlays contours for snow cover end dates on colour maps of end date trends. Snowmelt becomes independent of air temperature as exchange coefficients approach zero and independent of solar radiation as albedos approach 1. Lower albedos and larger exchange coefficients lead to earlier melt at Col de Porte, as might be expected. At Weissfluhjoch, however, low albedos can

cause radiation-driven melt in May, when solar radiation is high but air temperatures are still often below 0°C (Figure 1d); larger exchange coefficients then delay melt by cooling the snow, so the May and June contours curve downwards in Figure 4b. Even in the absence of net solar radiation and sensible heat ($\alpha = 1, C_H = 0$), there is sufficient longwave radiation in the driving data to melt the snow at Col de Porte each year, but the 2PM parameter space includes simulations that develop permanent





snow cover at Weissfluhjoch (upper left corner of Figure 4b) if the previous winter's snow has not melted by mid-August.

Average observed and ESM-SnowMIP model snow cover end dates at Col de Porte fall in April or May; 2PM can produce a wide range of end date trends for snow melting in those months, seen as a bulge in Figure 3a corresponding with a region where trend and end date contours cross in Figure 4a. ESM-SnowMIP models that have average end dates close to the start of May for Col de Porte have trends at the lower end of the 2PM range in Figure 3a, consistent with small exchange coefficients characteristic of low roughness and high atmospheric stability over snow.

Trends in snow cover end date show two areas of the 2PM parameter space in Figure 4 with enhanced negative trends. Strong trends for snow melting in July have already been noted in Figure 3 and will be discussed again later. Enhanced trends also occur for snow cover ending in months with warming trends, provided that exchange coefficients are large enough for simulations to be sensitive to air temperature. This is apparent in Figure 4 as protrusions of stronger trends along the end date contours for late April at Col de Porte and mid-June at Weissfluhjoch. The average end date of continuous snow cover

(dotted contour) is close to the date of maximum temperature trend (dashed contour) at Col de Porte, as expected for a positive feedback on warming with decreasing snow cover duration. Snow disappears at Weissfluhjoch about two weeks later than the date of maximum warming, however. It may be that warming trends at Weissfluhjoch are dominated by advection from lower surrounding areas with earlier snowmelt; Col de Porte is at the fifty-seventh elevation percentile and Weissfluhjoch is at the ninety-fourth elevation percentile for 10 km × 10 km areas centred on the sites. Warming is also expected to vary with elevation

in mountain regions (Pepin et al., 2015).

Annual snow cover duration (SCD) depends on the timing of snowfall, how much snow has to be melted and how much energy is available to melt it. Figure 5a for Weissfluhjoch and Table 2 for all sites show that modelled interannual variations in SCD are highly correlated with annual snowfall, except at Sodankylä; low snowfall and rapid temperature increases from April to May at Sodankylä limit variations in the end date of snow cover, both between years and between models (Figure

3c). Beyond the range of the ESM-SnowMIP models in Figure 5a, correlations in 2PM simulations inevitably decrease as the model undergoes a transition from seasonal to permanent snow cover at Weissfluhjoch independent of annual snowfall. Incoming solar radiation in the driving data for Weissfluhjoch peaks around the summer solstice in late June, whereas energy available to melt snow from longwave radiation and sensible heat peak in late July (measured solar radiation actually peaks in May at Weissfluhjoch because of seasonal variations in cloud cover and multiple reflections between high albedo snow and

clouds). Snow persisting after the peak in available energy will melt more slowly, so additional snowfall increases SCD more for simulations that retain seasonal snow cover later. The sensitivity obtained by linear regression of SCD against snowfall, shown for Weissfluhjoch in Figure 5b, therefore increases for late lying snow. Because SCD is highly correlated with snowfall, increased sensitivity to snowfall in simulations with late lying snow and decreasing snowfall combine to amplify trends in SCD, as seen in Figures 3 and 4 for both Col de Porte and Weissfluhjoch.



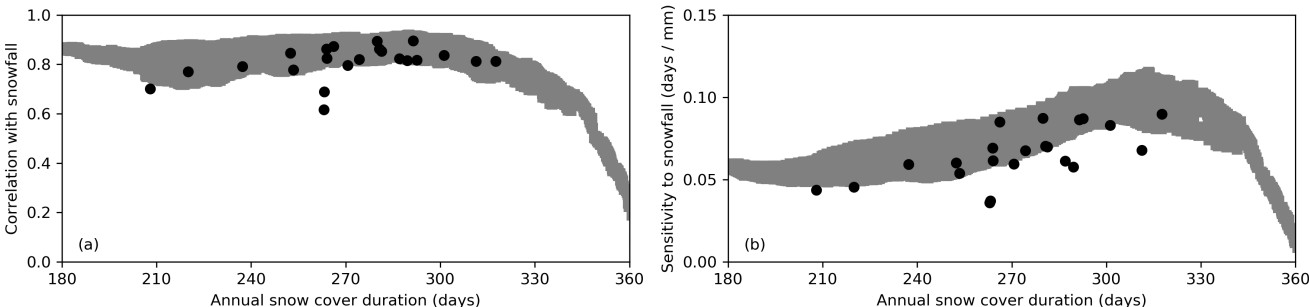

**Figure 5.** (a) Correlations between annual snow cover duration and snowfall amount in ESM-SnowMIP models (circles) and 2PM (grey band) at Weissfluhjoch. (b) Sensitivity of Weissfluhjoch snow cover duration to snowfall, expressed as increase in days of annual snow cover per mm of increase in annual snowfall found by linear regression.

**Table 2.** Average correlations between simulated annual snow cover duration and annual snowfall

| Site | ESM-SnowMIP | 2PM |
|---|---|---|
| Col de Porte | 0.79 | 0.81 |
| Reynolds Mountain East | 0.77 | 0.82 |
| Sodankylä | 0.50 | 0.45 |
| Weissfluhjoch | 0.81 | 0.79 |

## 170  4   Discussion and conclusions

Despite wide spreads in simulated snow cover durations, trends in models are consistent with observations at the ESM-SnowMIP sites: trends towards seasonal snow cover starting later in the year are not significant at any of the sites studied here, but there are significant trends towards seasonal snow cover ending earlier at Col de Porte and Weissfluhjoch (consistent with trends found across the Swiss Alps by Klein et al., 2016). Having been chosen for snow research in part because they

have dependable seasonal snow cover, the ESM-SnowMIP sites are not in regions of marginal snow cover that are most vulnerable to warming. A compilation of multiple observation-based estimates of Northern Hemisphere snow cover extent shows maximum decreasing trends in November and March, coincident with peaks in surface temperature warming trends (Mudryk et al., 2017). Large-scale simulations are required for predicting large-scale trends in snow cover extent, but simulations at well-instrumented sites allow more insight into modelling of snow processes and impacts that are experienced on small scales.

Interannual variations in modelled snow cover duration are strongly correlated with annual snowfall in the driving data at three of the four sites, which means that the models are also strongly correlated with each other (supplementary Figure A2) because they all shared the same driving data. This inter-model correlation will not be preserved when snow models are coupled to different atmosphere models. Coupling also allows feedbacks that are suppressed when snow models are driven with





prescribed meteorology. Coupled simulations with prescribed snow conditions are proposed in ESM-SnowMIP to evaluate the

effects of snow feedbacks (Krinner et al., 2018). Because water will not be conserved if snow mass is prescribed independently of snowfall and melt, these should be land-atmosphere simulations with prescribed sea surface temperatures to avoid perturbations of the ocean by runoff that would occur in coupled land-atmosphere-ocean simulations.

   A simple two-parameter snowmelt model shows that the response of snow models to warming depends on the timing of simulated melt in relation to the timing of warming and the strength of aerodynamic coupling between the surface and the

atmosphere. For simulations with snow cover persisting past mid-summer, responses to decreasing snowfall are amplified by increasing availability of energy as snow melts earlier. The same behaviour is observed in the spread of ESM-SnowMIP model snow cover end dates and trends for Weissfluhjoch; it should occur in reality for regions undergoing transitions from permanent to seasonal snow cover and on glaciers where the equilibrium line altitude is rising. This mechanism for amplification of snow climate sensitivity in addition to the well known snow albedo feedback has not been proposed before, as far as we are aware,

but it complements the "slower snowmelt in a warmer world" hypothesised by López-Moreno et al. (2013), Pomeroy et al. (2015) and Musselman et al. (2017) and observed on large scales by Wu et al. (2018) for snow melting in spring before the peak in available energy.

*Data availability.* The ESM-SnowMIP driving and evaluation data are available from https://doi.org/10.1594/PANGAEA.897575.

*Author contributions.* RE prepared the manuscript with substantial contributions from all co-authors. HK provided the global reanalysis

data, which LW extracted and interpolated for the study sites. All other co-authors either performed model simulations or provided field data.

*Competing interests.* The authors declare that they have no conflict of interest.

*Acknowledgements.* Analysis of the ESM-SnowMIP results was supported by NERC grant NE/P011926/1. Simulations were supported by the Russian Academy of Sciences Institute of Geography basic research program for SPONSOR (project No. 0148-2019-0009) and by the Russian Science Foundation for SWAP (Grant 16-17-10039).





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

## Appendix A: Supplementary figures

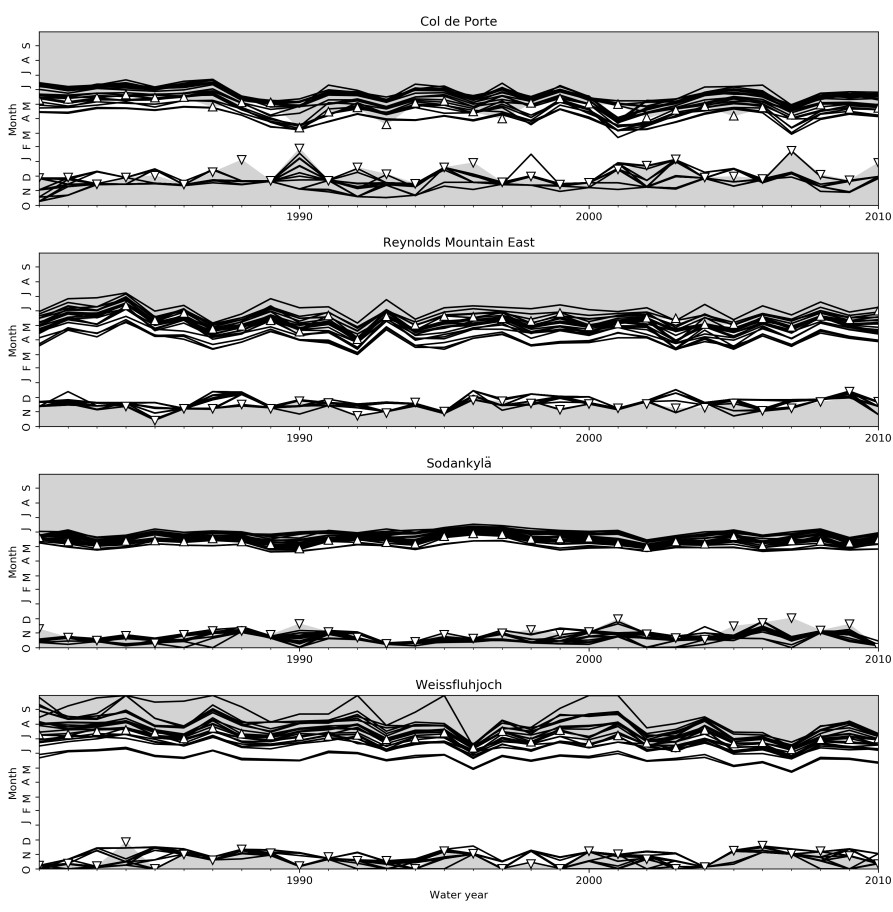

**Figure A1.** ESM-SnowMIP model predictions (lines) compared with observed start (▽) and end (△) dates of continuous seasonal snow cover at the sites. Snow-free periods are shaded (ephemeral summer snow cover is not shown).
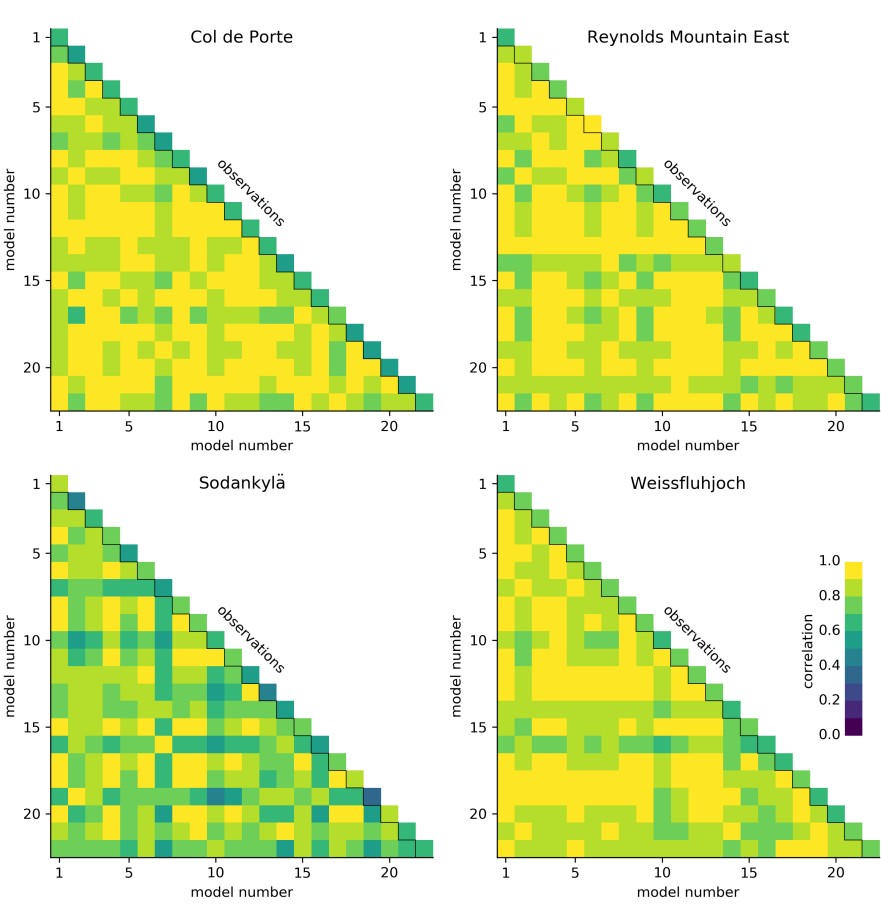

**Figure A2.** Correlations in interannual variations of snow cover duration between pairs of ESM-SnowMIP models. The upper diagonal in each figure shows correlations between individual models and observations at the sites.