# Peer review of "Snow cover duration trends observed at sites and predicted by multiple models"

_The Cryosphere, 2020_

## Referee Comment (RC1) · Anonymous Referee #1 · 4 Sep 2020

**Review of manuscript tc-2020-182**

**Summary**
This paper investigates snow cover duration and its temporal trend at 4 well-instrumented snow sites, using observations and a set of snow models which participate in the first phase of ESM-SnowMIP. To further understand the models' sensitivities at these sites, a simplified model depending only on two parameters is also used. The authors show that there is no statistically significant trend in snow cover start dates, whereas there is a significant trend in snow cover ending dates at two mountain sites. The simplified model shows that the magnitude of trends in snow end dates increases for simulations with end dates occurring at a later time in the year, and that a feedback mechanism exists between snow cover ending dates and the timing of warming. The same behaviour is found at one of the sites in the ESM-SnowMIP simulations.

I found the paper well written, with a clear and logical presentation of the methods and results. Most of the conclusions outlined are supported by the results presented (see below in the specific comments). The use of the simplified model allows to better understand the interplay between snow duration and trends, and their dependencies from the turbulent exchange coefficient and timing of warming. As a minor comment, I found a few sentences in the "Results" and "Conclusions" sections possibly difficult to read, and I encourage the authors to rephrase them (see specific comments below). Also, I appreciate the fact that only results that are statistically significant are fully interpreted by the authors, however reducing the number of sites used to draw some of the conclusions. In short, I think that the manuscript should be published after minor revisions.

**Specific comments**

Ln 127-128: I found this sentence a bit difficult to read. Could you reformulate it to make it clearer, as it seems an important point of the discussion?

Ln 162-165: This sentence is not backed up by evidence, or at least a reference to previous works, even though it is very reasonable. However, because it seems an important sentence to support the feedback argument explained by the authors in the following lines, I would suggest adding a plot (similar to Fig. 2) showing the annual cycle of the energy fluxes, or at least a reference to the literature.

Ln 188-190: I found this sentence too general and not specific enough. Could you please expand it, being more specific on the relationships between snow ending date, timing of warming and strength of aerodynamic coefficient, as found by the 2PM simulations?

Ln 188-197: A limitation of the present study is that these conclusions are derived mainly from one site. More sites, or global land-surface only simulations, are required to evaluate these conclusions on a larger range of climate conditions. I think the authors should state this aspect in a clearer way in the conclusions.

**Minor comments**

Ln 10: "should be able" → "are able"?

Ln 54: "atmosphere model" → "atmospheric model"?

Ln 54: Please add a reference to GSWP3 and the "Global Spectral Model"

Ln 71-72: sensible heat and moisture fluxes are between the surface and the atmosphere, not only from the surface to the atmosphere.

Ln 85: Please, add here how snow start and end dates are computed.

Ln 89: Are Lyon and Boise CRUTS locations as well?

Ln 143: "have trends at lower end" : is it the "higher end", right? Or "lower" in absolute value.

Ln 146-147: is this sentence implying a comparison with Fig. 2? Is so could you please state it in the text?

Ln 149: could you please specify that you are referring here to the observed snow cover?

Ln 172: I would clarify that only four of the ESM-SnowMIP sites are used in the study.

**Comments on figures**

Figure 5: is the quantity on the x-axis the mean annual snow cover duration for each model?

---

## Referee Comment (RC2) · Anonymous Referee #2 · 8 Sep 2020

Summary:

The authors present an evaluation of 30-year trends in simulated snow-cover duration metrics. In-situ data from four well-instrumented sites are used as validation of multiple snow models forced with bias-corrected, gridded meteorological data. A simplified snow model consisting of two main parameters is used to provide process-level insight into the simulated snow sensitivity to climate. The authors report limited (statistically significant) trends in snow-cover onset and more notable declines in the date of snow-cover depletion. The simplified model is used to demonstrate that regions most sensitive to the influence of warming are those where snow historically persists late into the spring and summer.

The paper is well written and easy to follow. The topic is clearly germane to the scope

of The Cryosphere. The focus on snow-cover duration metrics (start-date, end-date, and days with snow-cover) makes the results relevant to a wide audience (ecologists, hydrologists, climate scientists). I have some questions about the exact definition of these metrics and sensitivity of results (see general comments). I especially like the use of the simplified model and the fact that it generally spans the response of the multi-model ensemble; nimbly side-stepping the vague inference commonly required to describe the varied responses of very complex models. I support publication, but am requesting a re-analysis of the model output regarding how snow-cover metrics are calculated.

General Comments:

1) The authors use the term "continuous seasonal snow-cover" to describe start and end dates. I agree that snow-cover start- and end-dates should be based on "continuous" seasonal snow data; however, the authors' definitions are at odds with this. Line 109 states "Start and end dates for continuous seasonal snow cover were found by searching for the first and last dates with snow depths exceeding 2 cm before and after the dates of maximum snow depth in each year."

Trujillo and Molotch (2014) define continuous seasonal snow-cover start and end dates as the last start date before maximum accumulation and the first snow-disappearance after maximum accumulation. This definition limits assessment to seasonal snow-cover and excludes early and late snow accumulation events that would arguably be 1) short-lived, 2) possible synoptic outliers, and 3) potentially of limited hydrological or climatological significance. My concern is that >2 cm of snow may accumulate in September or June, and last only a few days, while the true measure of continuous seasonal snow-cover may persist from November to May. Please ensure that your analysis focuses on continuous seasonal snow-cover as stated in the paper rather than the methods described on Lines 109-110.

2) How are the conclusions stated on line 135 (radiation-driven melt occurring when

air temps are below 0C) impacted by the underlying assumption of 2PM that ignores snowpack cold content? Is this consistent with observations? Is it consistent with models that include representation of cold content dynamics? For example, according to López-Moreno et al (2013) referenced in this paper: "With cold temperatures, solar radiation at the end of winter is not sufficient to trigger melting, ..."

3) Many methodological descriptions are included in the Results section. Please consider moving to the Methods section.

Specific Comments:

Line 12: What does it mean to "demonstrate climate change"? Be more explicit.

Line 13: "Reports from the second onwards have. . ." should be rewritten for grammar.

Line 195: The final sentence of the paper seems to dispute the origin of a "slower snowmelt in a warmer world" hypothesis stated as the title of a 2017 paper by Musselman et al. A quick scan of the López-Moreno et al (2013) paper doesn't include this messaging or the words "slow", "world" or "hypothesis". Pomeroy et al (2015) is a conference proceeding not accessible at the link provided. The work of Musselman et al (2017) was built upon graduate student work published in

Musselman, K. N., Molotch, N. P., Margulis, S. A., Kirchner, P. B., & Bales, R. C. (2012). Influence of canopy structure and direct beam solar irradiance on snowmelt rates in a mixed conifer forest. Agricultural and Forest Meteorology, 161, 46-56.

Page 53 of that paper: "Reduced ablation rates were observed at lower elevations where snowmelt commences earlier in the year when solar elevations are lower. At upper elevations snowmelt continues later into the year when solar elevations are higher, resulting in greater seasonal ablation rates."

Please either be complete and accurate in describing the origin, or succinctly cite the paper stating the referenced hypothesis.

---

## Author Comment (AC1) · 21 Oct 2020

Ln 127-128: I found this sentence a bit difficult to read. Could you reformulate it to make it clearer, as it seems an important point of the discussion?

Rewritten as "2PM simulations that melt snow later in the year at Col de Porte and Weissfluhjoch have stronger negative trends in continuous snow cover end dates".

Ln 162-165: This sentence is not backed up by evidence, or at least a reference to previous works, even though it is very reasonable. However, because it seems an important sentence to support the feedback argument explained by the authors in the following lines, I would suggest adding a plot (similar to Fig. 2) showing the annual cycle of the energy fluxes, or at least a reference to the literature.

A new figure showing annual cycles of energy fluxes (number 2 in the revised manuscript) and a new paragraph (lines 99 to 106) have been added to address this comment.

Ln 188-190: I found this sentence too general and not specific enough. Could you please expand it, being more specific on the relationships between snow ending date, timing of warming and strength of aerodynamic coefficient, as found by the 2PM simulations?

Rewritten to be more specific as "A simple two-parameter snowmelt model shows that the response of snow models to warming in their driving data is stronger in simulations that melt snow close to the time of year when the warming is strongest and in simulations with stronger aerodynamic coupling between the surface and the atmosphere".

Ln 188-197: A limitation of the present study is that these conclusions are derived mainly from one site. More sites, or global land-surface only simulations, are required to evaluate these conclusions on a larger range of climate conditions. I think the authors should state this aspect in a clearer way in the conclusions.

This is an important point on which to end the paper, with the addition of "Conclusions drawn here have been based on simulations at a limited number of sites. The global land-only simulations now being performed for LS3MIP (van den Hurk et al. 2016) will provide an opportunity for testing these conclusions in a much wider range of climate conditions."

Minor comments

Ln 10: "should be able" → "are able"?

"are able" claims an ability of global climate models that is not demonstrated here.

Ln 54: "atmosphere model" → "atmospheric model"?

Done

Ln 54: Please add a reference to GSWP3 and the "Global Spectral Model"

References to Kim (2017) for the GSWP3 project and Yoshimura and Kanamitsu (2008) for the global spectral model downscaling have been added.

Ln 71-72: sensible heat and moisture fluxes are between the surface and the atmosphere, not only from the surface to the atmosphere.

Now written as "between the surface and the atmosphere"

Ln 85: Please, add here how snow start and end dates are computed.

Added "with depths exceeding 2 cm" to explain.

Ln 89: Are Lyon and Boise CRUTS locations as well?

Yes. This is now stated.

Ln 143: "have trends at lower end": is it the "higher end", right? Or "lower" in absolute value.

Rephrased as "trends at the less negative end".

Ln 146-147: is this sentence implying a comparison with Fig. 2? Is so could you please state it in the text?

It is actually referring to Figure 1, which is now stated in the text.

Ln 149: could you please specify that you are referring here to the observed snow cover?

Done

Ln 172: I would clarify that only four of the ESM-SnowMIP sites are used in the study.

Done

Figure 5: is the quantity on the x-axis the mean annual snow cover duration for each model?

Yes. The axis label has been changed to "Average annual snow cover duration (days)".

**Anonymous Referee #2**

General Comments:

1) The authors use the term "continuous seasonal snow-cover" to describe start and end dates. I agree that snow-cover start- and end-dates should be based on "continuous" seasonal snow data; however, the authors' definitions are at odds with this. Line 109 states "Start and end dates for continuous seasonal snow cover were found by searching for the first and last dates with snow depths exceeding 2 cm before and after the dates of maximum snow depth in each year." Trujillo and Molotch (2014) define continuous seasonal snow-cover start and end dates as the last start date before maximum accumulation and the first snow-disappearance after maximum accumulation. This definition limits assessment to seasonal snow-cover and excludes early and late snow accumulation events that would arguably be 1) shortlived, 2) possible synoptic outliers, and 3) potentially of limited hydrological or climatological significance. My concern is that >2 cm of snow may accumulate in September or June, and last only a few days, while the true measure of continuous seasonal snowcover may persist from November to May. Please ensure that your analysis focuses on continuous seasonal snow-cover as stated in the paper rather than the methods described on Lines 109-110.

The definition used here is the same as in Trujillo and Molotch (2014) and many other studies. To clarify, the description has been rephrased as "Start and end dates for seasonal snow cover were found by searching for the last date with snow depths less than 2 cm before the maximum snow depth in each year and the first such date after the maximum."

2) How are the conclusions stated on line 135 (radiation-driven melt occurring when air temps are below 0C) impacted by the underlying assumption of 2PM that ignores snowpack cold content? Is this consistent with observations? Is it consistent with models that include representation of cold content dynamics? For example, according to López-Moreno et al (2013) referenced in this paper: "With cold temperatures, solar radiation at the end of winter is not sufficient to trigger melting, ..."

Snow melt is determined by energy balance, not air temperature. Melt at sub-zero temperatures (and subsurface melt while the snow surface is still frozen) is certainly possible. However, the statement referred to in this comment is the explanation of behaviour seen in a region of the parameter space that is implausible for pristine snow. "low albedos" has been replaced with "low 2PM albedos" to emphasize that it is model behaviour that is being discussed. To put the 2PM assumption into context, the information that "cold content of snow is represented in more sophisticated models" and "21 kJ will warm 1 kg of snow from -10°C to 0°C but will only melt 63 g of snow at 0°C" have been added to the model description.

3) Many methodological descriptions are included in the Results section. Please consider moving to the Methods section.

The sentence describing the number of 2PM simulations and parameter ranges has been moved from Results to Methodology. Retaining some other small methodological comments aids understanding of the results section.

Specific Comments:

Line 12: What does it mean to "demonstrate climate change"? Be more explicit.

Rewritten as "evidence for climate change".

Line 13: "Reports from the second onwards have..." should be rewritten for grammar.

This sentence was grammatically correct but unnecessarily convoluted. It has been rewritten as "Strong relationships between decreasing snow cover and increasing air temperature have been demonstrated in observations and in multi-model simulations for the Coupled Model Intercomparison Project (CMIP)".

Line 195: The final sentence of the paper seems to dispute the origin of a "slower snowmelt in a warmer world" hypothesis stated as the title of a 2017 paper by Musselman et al. A quick scan of the López-Moreno et al (2013) paper doesn't include this messaging or the words "slow", "world" or "hypothesis". Pomeroy et al (2015) is a conference proceeding not accessible at the link provided. The work of Musselman et al (2017) was built upon graduate student work published in Musselman, K. N., Molotch, N. P., Margulis, S. A., Kirchner, P. B., & Bales, R. C. (2012). Influence of canopy structure and direct beam solar irradiance on snowmelt rates in a mixed conifer forest. Agricultural and Forest Meteorology, 161, 46-56. Page 53 of that paper: "Reduced ablation rates were observed at lower elevations where snowmelt commences earlier in the year when solar elevations are lower. At upper elevations snowmelt continues later into the year when solar elevations are higher, resulting in greater seasonal ablation rates." Please either be complete and accurate in describing the origin, or succinctly cite the paper stating the referenced hypothesis.

The impression of a dispute has been removed. The abstract of López-Moreno et al (2013) states that "Melting rates decreased with increased temperature", but their discussion states the corollary that "melting rates tended to increase under colder conditions". "Slower snowmelt in a warmer world" is, indeed, quoting the title of Musselman et al. (2017). For clarity, the López-Moreno and Musselman references are now dealt with in separate sentences:
"This mechanism for amplification of snow climate sensitivity in addition to the well known snow albedo feedback has not been proposed before, as far as we are aware, but it complements the `slower snowmelt in a warmer world' hypothesised by Musselman e al. (2017} and observed on large scales by Wu et al. (2018} for snow melting in spring before the peak in available energy. López-

Moreno et al (2013) found accelerated melt rates in simulations with colder temperatures that delayed the start of melt until later dates with more intense solar radiation."